# A Lightweight Subgraph-Based Deep Learning Approach for Fall Recognition

**DOI:** 10.3390/s22155482

**Published:** 2022-07-22

**Authors:** Zhenxiao Zhao, Lei Zhang, Huiliang Shang

**Affiliations:** 1School of Information Science and Technology, Fudan University, Shanghai 200433, China; 19210720040@fudan.edu.cn; 2Academy for Engineering and Technology, Fudan University, Shanghai 200433, China; leizhang18@fudan.edu.cn

**Keywords:** fall recognition, skeleton extraction, deep learning, sub-graph

## Abstract

Falls pose a great danger to social development, especially to the elderly population. When a fall occurs, the body’s center of gravity moves from a high position to a low position, and the magnitude of change varies among body parts. Most existing fall recognition methods based on deep learning have not yet considered the differences between the movement and the change in amplitude of each body part. Besides, some problems exist such as complicated design, slow detection speed, and lack of timeliness. To alleviate these problems, a lightweight subgraph-based deep learning method utilizing skeleton information for fall recognition is proposed in this paper. The skeleton information of the human body is extracted by OpenPose, and an end-to-end lightweight subgraph-based network is designed. Sub-graph division and sub-graph attention modules are introduced to add a larger perceptual field while maintaining its lightweight characteristics. A multi-scale temporal convolution module is also designed to extract and fuse multi-scale temporal features, which enriches the feature representation. The proposed method is evaluated on a partial fall dataset collected in NTU and on two public datasets, and outperforms existing methods. It indicates that the proposed method is accurate and lightweight, which means it is suitable for real-time detection and rapid response to falls.

## 1. Introduction

As a part of abnormal action recognition, fall recognition holds great significance. Fall recognition is an important branch in the medical and health care field. It can not only reduce the reliance on manual labor, but also provide technical support for the follow-up process such as medical alarm systems, intelligent monitoring, and smart homes. Therefore, fall recognition has a broad application prospect and great economic value.

Population aging is mainly caused by the decline in the birth rate and increase in life expectancy, which has become a matter of concern [1]. According to statistics [2], the global elderly population aged 65 or over has reached 727 million in 2020, and it will probably be 1.5 billion by 2050. In 2020, the proportion of elderly people in this age group was 9.3%, and it will reach 16.0% in 2050. The most likely outcome of falls is fractures, which can lead to physical disability, which hinders the ability to live independently, as well as a psychological fear of falling down again. Falls can cause not only moderate or severe injuries to older adults, but also an emotional burden and financial stress for them and their relatives. In the face of this situation, it is particularly important to recognize falls in elderly people quickly and effectively. On one hand, it can alert the guardians for help in time to mitigate the injuries caused by falls; on the other hand, it can save public medical resources and reduce the public medical burden on society.

Generally, falls can be recognized by several methods, including wearable sensors, traditional geometric and motion feature detection, and deep learning methods [3]. The first two methods have some challenges, such as frequent manual calibration, short battery life, and high cost, while the methods based on deep learning are more feasible and robust. However, existing fall recognition methods based on deep learning often fail to detect falls well owing to the lack of complete data and complex network extraction of fall features, which are prone to overfitting in training. In the fall recognition system, a timely response is very important, and even a small delay may lead to huge losses. It is necessary to reduce the computational complexity and improve the system response. Therefore, designing a lightweight and fast vision-based method for automatic fall recognition is an important task.

As shown in Figure 1, when a fall occurs, if we consider the whole human body as a point, the direction of the point and the law of change are different when making different movements. The body’s center of gravity moves from a high position to a low position, and the spatial position of the person changes. The magnitude of the change varies among body parts. The characteristic information specific to each type of movement is hidden in these differences. However, similar to falls, the center of gravity changes when sitting, squatting, and so on are performed, so if fall recognition is based only on the overall change in the gravity of the skeleton, this may be confusing and lack sufficient directionality. After observing and studying a large number of falls, it is found that there are significant differences in the alignment and magnitude of changes in human body parts during falls. For example, when a person falls, the change in the center of gravity of the head is much greater than that of the arms and legs, so it is important to consider the differences in the characteristics of specific parts rather than just the whole body.

In this paper, an end-to-end lightweight subgraph-based deep learning method is proposed to simultaneously extract and fuse joint information to obtain richer spatial-temporal contextual information, which provides more clues for feature representation of skeletal sequences. Firstly, we use the OpenPose [4] skeleton extraction algorithm to obtain the skeleton data of a human body. The skeleton data are processed and fused, and the semantic guidance of types is added respectively. Based on this, adaptive graph convolution and sub-graph attention modules are introduced to emphasize the more important parts of the spatial-temporal joints and skeleton, allowing the model to add a larger perceptual field while maintaining its lightweight characteristics. Then, a multi-scale temporal convolution module is also designed at the same time to extract and fuse multi-scale temporal features using different sizes of perceptual fields, which enriches the feature representation.

In summary, the major contributions of this work are as follows:(1)Based on the skeleton data, we propose an end-to-end lightweight subgraph-based deep learning method that achieves better recognition accuracy while ensuring a lower number of parameters.(2)A sub-graph division model is introduced, and a subgraph-based attention module is embedded to achieve better feature representation capability for fall behavior.(3)A multi-scale temporal convolution module is introduced in our model to enhance temporal feature representation.(4)A total of six categories of falls and behaviors similar to falls are collected in the NTU [5] dataset, and skeleton extraction was performed using OpenPose on two publicly available fall behavior datasets, i.e., UR Fall Detection Dataset [6] and UP-Fall detection dataset [7]. The model is validated in these three datasets.

The rest of this paper is organized as follows. Related work is reviewed in Section 2, and Section 3 explains the proposed method. In Section 4, we present three datasets, as well as the experimental results and discussion, and we provide a conclusion in Section 5.

## 2. Related Work

### 2.1. Skeleton-Based Action Recognition

Generally, deep learning methods based on the human skeleton include three main categories: recurrent neural network (RNN), convolutional neural network (CNN), and graph convolutional network (GCN). Owing to the compactness and robustness compared with RGB-based representations, skeleton-based action recognition is receiving increasing attention. Previous methods usually converted the original skeleton data into fixed forms, such as point sequences or pseudo images. Then, they can be input into RNN or CNN and other deep networks for feature extraction [8,9,10,11,12,13,14]. Driven by the development of the graph, Yan et al. [15] proposed spatial temporal graph convolutional networks (ST-GCNs) to model the spatial information and temporal motion information of the skeleton, so as to construct the skeleton spatial-temporal map. Accordingly, a series of GCN-based methods have emerged [16,17,18]. They utilize the topological structure of human skeleton to aggregate features of relevant skeleton nodes and temporal sequences. Therefore, the GCN-based method shows better performance and accuracy compared with previous methods.

### 2.2. Subgraph-Based Methods

In the subgraph-based approaches, the human skeleton is divided into several parts according to the human structure, and the features of each part are extracted separately. Thakkar et al. [19] divided the human skeleton graph into five sub-graphs, which share some nodes. They used GCN operations in the sub-graphs, and then spread features between sub-graphs through shared nodes. Huang et al. [20] used subgraph-based GCN to obtain discriminative information of joints and body parts. These subgraph-based approaches employ sophisticated strategies to disseminate information individually or integrate information from all parts, thereby reducing computational costs.

### 2.3. Fall Recognition

Depending on the device and the method used, fall recognition algorithms are divided into three main areas: wearable sensor-based methods, scene-based methods, and vision-based methods [3]. Wearable sensor-based approaches have many advantages because they use sensors to automatically detect falls and send help-seeking messages to health care workers through communication devices such as WIFI, mobile networks, and Bluetooth [21,22,23,24,25]. However, elderly people often forget to wear sensors. Wearable sensors capture abnormal values, such as velocities and angles, and send alerts to users. Wearable sensors require frequent charging, which can make it difficult for older adults to continuously monitor battery charge status. Wearable sensors are also less comfortable. This option is not well suited for the elderly because of frequent charging, discomfort, and other side effects of sensors.

Scene sensor-based approaches use scene sensors installed in the monitored area to collect information such as pressure, vibration, and sound to determine whether a fall has occurred. However, such sensors are sensitive to noise information and are prone to false alarms [26,27]; furthermore, the arrangement cost is high and the accuracy rate is low. Computer vision-based fall detection methods detect whether a fall has occurred by passively acquiring human motion information from the monitoring device and processing the acquired video or image. The vision-based method does not require the user to wear any device and has a good user experience and high detection accuracy [28].

## 3. Method

The dynamic changes of the human skeleton provide important information for human movement recognition and joint displacement, which are a direct reflection of human movement. The fall sequence of the human skeleton is an important source of human fall information. In this paper, we study the features of human skeleton fall sequences rather than traditional features.

For the task of fall recognition, the modeling of spatial-temporal information and multi-scale contextual information is crucial for the final classification and prediction results. To meet the requirements of both accuracy and efficiency, in this chapter, an end-to-end network is designed, as shown in Figure 2. The design adopts a single-stream structure, which separates the spatial modeling and temporal modeling, which is conducive to the expression of spatio-temporal information of the skeleton, and reduces the complexity of processing. Firstly, skeleton data are obtained by OpenPose, then multiple information flow data of joints are fused and encoded. Secondly, through the spatial module, the semantics of types are added to joints and the whole graph is divided into several sub-graphs, and the spatial information is modeled by the adaptive graph convolution module that introduces sub-graph attention. Then, it is input to the temporal module, which is used to aggregate information between different nodes in different frames, through spatial max-pooling (SMP) and temporal max-pooling (TMP). A multi-scale temporal convolution network (MTCN) is introduced to fully extract the temporal contextual information. Finally, the final recognition score is obtained by the softmax function through the fully connected layer.

### 3.1. Feature Encoding

The human skeleton contains not only data of joint points, but also multi-modal data such as joint motion information flow. The full fusion of such multi-modal data can ensure the construction of a globally adaptive adjacency matrix, which in turn reduces the computational cost and model complexity and improves the computational efficiency.

For a given skeleton sequence, the input sequence is represented as $\;X=\left\{x\in R^{C_{in}*T_{in}*V_{in}}\right\}$, where $C_{in},\;T_{in},\;V_{in}$ are the coordinate dimension, the frame dimension, and the joint dimension, respectively. The joint node is defined as $s=\{V_{i,t}|i=1,2,\ldots ,N;t=1,2,\ldots ,T\}$, where *n* is the number of joints within the skeleton and T is the total number of frames in the sequence. The relative position coordinates $r_{i,t}$=$V_{i,t}-V_{c,t}$, where *c* is denoted as the position of the central node. The relative position and absolute position are concatenated to obtain the position coordinates of the joint.

The first-order motion and second-order motion of the joints are expressed as
(1)$$f_{i,t}=V_{i,t+1}-V_{i,t}$$
(2)$$s_{i,t}=V_{i,t+2}-V_{i,t}\;.$$

The first-order flow $f_{i,t}$ and the second-order flow $s_{i,t}$ are concatenated to obtain the joint motion. An ordinary 1 × 1 convolutional layer and a Relu layer are used to encode the input features to obtain the high-dimensional expressions ${\bar{J}}_{t,i}\in R^{C_{1}}$ and ${\bar{m}}_{t,i}\in R^{C_{1}}$, and here $C_{1}=16$. Thus, the feature encoding form of the joint is expressed as
(3)$$E_{t,i}={\bar{J}}_{t,i}+{\bar{m}}_{t,i}\;,\;\in R^{C_{1}}\;.$$

We use the semantics of joint types and the encoded features of the input to learn the connection dependencies between the nodes in the frame, and the semantic information of joint types facilitates the learning of a more appropriate adjacency matrix. The joint type semantics of the i-th node in the graph is noted as $p_{i}\in R^{V_{0}}$, encoded using one-hot vectors with the i-th position being 1 and the remaining positions filled with 0 elements. A two-layer fully connected layer yields the encoding of the k-th node type semantics in the form.
(4)$${\bar{p}}_{i}=\;\sigma \left(W_{2}\left(\sigma \left(W_{1}p_{i}+b_{1}\right)\right)+b_{2}\right),\in R^{C_{1}},$$
where $W_{1}$ and $W_{2}$ are the weight matrices of the fully connected layer and σ is the Relu function. Then, the k-th joint node is jointly represented in time sequence as $E_{t,i}=\left[E_{t,i},{\bar{p}}_{i}\right]\;\in R^{2C_{1}}$.

### 3.2. Adaptive Graph Convolution Networks

Each adaptive graph convolution consists of a standard spatial graph convolution layer (SGCN), a standard temporal convolution layer with a kernel size of $K_{t}*1$, and a sub-graph attention (SGA) module, as shown in Figure 3. Inspired by DGNN [29], the adjacency matrix A of SCGN is made to be fixed in the first 10 epochs of training and learnable as model parameters in the later stages of training, thus facilitating network training and enhancing more flexibility to achieve adaptive effects. Three such adaptive graph convolution modules are stacked, and the number of module output channels for each adaptive graph convolution is 64, 128, and 256, respectively.

**Sub-Graph Division.** The human skeleton is a natural topological map consisting of five main body parts, and there are distinct differences in each part when falls occur. The skeleton map is divided into several sub-graphs, usually five subgraphs. Each subgraph represents a part of the natural structure of the human body, i.e., the torso, hands, and legs. Each part is naturally connected and shares joint points between adjacent parts, thus covering all natural connections between joints in the skeleton diagram. For example, between the torso and left arm, the shoulder joints overlap. The skeleton diagram G is divided into five parts, and the division process is shown as
(5)$$G=\underset{p\epsilon \left\{1,\ldots g\right\}}{\sum }G_{p}|G_{p}=(V_{p},E_{p})$$
where $G_{p}$ is a subgraph of the skeleton graph G. The set of vertices in this subgraph is $V_{p}$ and the set of edges is $E_{p}$. Treating each subgraph as an independent graph, the subgraphs are all part of the overall skeleton graph, and the overall skeleton graph must contain the nodes of each subgraph. Therefore, the adjacency matrix *A* of the whole graph can be represented by the adjacency matrix of the subgraphs.
(6)$$A=\underset{l=0,1,2}{\sum }A_{l}$$
(7)$$A_{l}=\underset{p}{\sum }A_{pl}$$
where $p\in \left\{1,\;\ldots \;g\right\}$, l = 3, l is the sampling strategy. $A_{pl}$ denotes the adjacency matrix corresponding to the *l*-th adjacency subset of the *p*-th subgraph, and $A_{pl}$ is set to a V × V matrix, with V being the number of nodes of the human skeleton graph. The adjacency matrix of each sub-graph is superimposed to obtain each adjacency subset, and the three adjacency subsets are stitched to obtain the adjacency matrix *A* of the full graph, which is represented as an l × V × V dimensional tensor containing the dependencies of each sub-graph and the connections between joints.

**Sub-Graph Attention (SGA).** In falls, the importance of different body parts in the whole action sequence is different, so it is necessary to give each body part a different attention score. As shown in the Figure 4, the five sub-graphs are obtained by picking the corresponding nodes.
$$X_{J}\to \left[x_{1},x_{2},\ldots ,x_{P}\right],\;x_{i}\in R^{Co\times T\times M},\;X_{J}\in R^{Co\times T\times V}$$
where *M* denotes the maximum number of nodes in each sub-graph, and parts less than *M* are filled with 0. Each part is stitched together after averaging pooling at the frame level, and the number of channels is reduced to 1/4 of the original through a fully connected layer. After a batch normalization layer, a sigmoid non-linear layer, it is separated into five sub-vectors. Then, the channel number dimension is reduced through a linear layer, and softmax is solved for each sub-graph part. Finally, weights are reassigned to obtain an attentional feature map based on the sub-graph. The formula is expressed as follows.
(8)$$f_{out}=concat\{f\left(p\right)⊗\theta \left(\sigma \left(\mathrm{pool}(f_{p}\right)W+b\right)W^{\prime }+b^{\prime })\},\;p=1,2,\ldots ,5,$$
where ⊗ denotes the array elements multiplied sequentially, *θ* is the softmax function, and *σ* is the sigmoid function.

### 3.3. Multi-Scale Temporal Convolution Network (MTCN)

In the temporal module, spatial max-pooling (SMP) and temporal max-pooling (TMP) are used to aggregate information between different nodes on different frames. In temporal modeling, previous works often use one-dimensional convolution to extract motion features. Considering that similar actions are distinguished by fine-grained information and slowly changing motions span across multiple consecutive frames, we design a multi-scale temporal convolution network (MTCN) to extract features at different time scales. Specifically, different time scale branches are processed by temporal convolution with convolution kernel sizes of 3 × 1, 5 × 1, and 7 × 1 in the time dimension, respectively. Then, the features generated by the three branches are fused at different stages. The multi-scale temporal convolution facilitates the extraction of richer contextual information, which improves the feature representation capability of the model. The output is represented as
(9)$$f_{out}=F_{3,1}(f_{in})⊕F_{5,1}(f_{in})⊕F_{7,1}\left(f_{in}\right).$$

Finally, our network is input to the fully connected layer and the final fall recognition score is obtained by softmax.

## 4. Experiments

### 4.1. Datasets and Evaluation Measures

To verify the performance of the proposed method, we evaluated it on several public datasets, including the NTU dataset, UR Fall Detection Dataset (URFD), and UP-Fall detection dataset. We collected a part of fall data and non-fall data in NTU RGB and extracted the skeleton of the URFD and UP-FALL detection dataset using the OpenPose algorithm to obtain the corresponding 3D skeleton data.

The NTU dataset was captured simultaneously by 3 Kinect v2 cameras. It contains 60 types of actions, usually also known as NTU 60. In this dataset, the proposed method is usually evaluated using two protocols: cross-subject (X-sub) and cross-view (X-view). The X-sub contains 40,320 training samples and 16,560 test samples, and the partitioning rule is based on 40 subjects, indicating that the behaviors in the training and test sets are from different actors. The X-view takes the ones collected by camera 2 and 3 as the training set (37,920 samples) and those collected by camera 1 as the test set (18,960 samples). The names and numbers of specific nodes correspond to each other, e.g., 3 for neck, 16 for left foot, 25 for right thumb, and so on. We collected some fall datasets from NTU and selected the fall behaviors and fall-like behaviors as the sample data, containing 948 items and 4506 items, respectively, including drop, pick up, sit down, wear a shoe, take off a shoe, and fall. We refer to the dataset consisting of these six types of actions as the NTU fall dataset. When the other five actions occur, such as drop and sit, the overall center of gravity of the body changes, which is similar to fall behavior, has a certain interference in the fall recognition task, and improves the recognition ability of the model.

The UR Fall Detection Dataset is a collection of human activity sequences captured by outdoor cameras installed at different viewpoints, including 30 falls and 40 video clips of daily living activities. In this paper, we only use RGB images of 30 fall sequences and 30 normal activity sequences. We used the OpenPose algorithm to extract the skeleton of this pair of datasets.

The UP-Fall datasets are collected from 17 young healthy subjects during 11 activities (5 falls and 6 daily activities) in 3 trials, including two datasets, i.e., a sensor-based dataset and a vision-based dataset. Especially, this dataset does not reflect the reality of a fall of an elderly person. The vision-based dataset is collected by fixing a front camera and a lateral camera in the room, which provides a front view and side view of the subject, respectively. In the proposed method, we only utilize the lateral camera view, which in turn reduces the computational complexity. The experiment proves it is sufficient to reach the comparable performance compared with the state-of-the-art methods using two camera angles. Similar to URFD, OpenPose is utilized to extract the skeleton information of the UP-Fall detection dataset.

To evaluate the performance of the proposed method, some metrics like sensitivity (recall), specificity, and model accuracy are calculated, which are shown as follows.
(10)$$\mathrm{sensitivity}=\frac{TP}{TP+FN}$$
(11)$$\mathrm{specificity}=\frac{TN}{TN+FP}\;$$
(12)$$\mathrm{accuracy}=\frac{TP+TN}{TP+TN+FP+FN}$$

In the above equations, *TP* means the number of falls that have been predicted as fall correctly, *FP* means the number of activities of daily living (ADL) that are predicted as falls, and *TN* means the number of normal activities that are predicted as normal activities. Similarly, *FN* means predicting a normal activity as a fall.

### 4.2. Implementation Details

All experiences are performed on the Pytorch platform for model training and optimization on machines with an NVIDIA GeForce GTX 2080Ti GPU. Training is performed using a stochastic gradient descent (SGD) optimizer with a Nesterov momentum of 0.9 and a weight decay of 0.0005, with cross entropy as the loss function. The training time is set to 60 epochs with an initial learning rate of 0.1. The learning rate decays by a factor of 10 at the 20th epoch and the 40th epoch. The size of all batches is set to 16. All skeleton sequences are filled to 300 frames by replaying the actions. In training, we perform data augmentation by rotating the 3D skeleton to some extent to simulate the change of camera viewpoint, and thus be more robust to view changes. In order to unify the data distribution, reduce the influence of the central position of the body in the image, and simplify the model training process, coordinate data normalization is adopted. For each sequence, we consider the central skeleton point of the first body in the first frame as the origin of the coordinate system and subtract the coordinates of the central point from the coordinates of each joint point to normalize the other frames to this coordinate system.

### 4.3. Results’ Comparison

We tested the proposed method and other classical methods on the UR Fall dataset. These methods are divided into two groups. The first two groups of methods are designed based on hand-crafted features [30,31,32] and the second group of methods employs deep learning [33,34].

Table 1 provides the comparison results of the proposed method and the other five methods on the UR Fall dataset. It can be seen from Table 1 that, compared with the other four methods, the proposed method has the best accuracy. Our proposed method fully leverages skeleton information and does not require handcrafted features, which significantly improves the performance of fall recognition. This work also performs better with the help of the GCN network and sub-graph division compared with the CNN-based methods [33,34]. The GCN network helps capture spatial-temporal information of the human body and the sub-graph attention helps enhance the feature representativeness, which improves the performance of the fall recognition task.

We also tested the proposed method on the UP-Fall dataset and compared it with several existing methods, as shown in Table 2. In method [35], the author conducts a case study on the lateral and frontal cameras respectively. We can see that, in our proposed method, all of the evaluation parameters, i.e., sensitivity, specificity, and accuracy, are relatively good. The author in method [7] applies some machine learning algorithms to the UP-Fall datasets, such as random forest (RF), support vector machine (SVM), multilayer perceptron (MLP), K-nearest neighbours (KNN), and CNN. Compared with the machine learning algorithm, our proposed method displays very high performance. The method [36] also uses CNN on the UP-Fall dataset, but compared with our method, it has achieved significantly lower accuracy. The method [37] also adopts RF, SVM, MLP, KNN, and other machine learning algorithms on the dataset using data from two camera inputs. However, our proposed method achieves similar accuracy only utilizing input from a single lateral camera. Based on this fact, it can be inferred that the proposed method is very effective. The method [38] employs CNN and LSTM on the UP-Fall dataset on the lateral camera, but compared with the proposed method, it has achieved lower sensitivity, specificity, and accuracy. Therefore, our proposed method can achieve similar or even better results than those works that use two or only one camera input to identify falls.

We also verified the performance of the proposed method on our collected NTU dataset, which consists of six types of actions. The sensitivity, specificity, and accuracy are 97.5%, 89.6%, and 94.5%, respectively, as shown in Table 3. In addition, we also draw a confusion matrix, including five behaviors that identify falls and are similar to falls, as shown in Figure 5. It can be seen from the figure that the methods in this chapter can better distinguish six actions on the NTU dataset, in which fall and drop are the most likely to be confused. For non-fall behaviors, such as put on a shoe and sit down, special analysis is needed. We analyze that the causes of false discrimination are as follows: (a) In pose estimation, the lack of joint points leads to data incompleteness, which reduces the final recognition results. (b) During the experiment, there is still a difference between real falls and recorded falls owing to the self-protection of the experimenter.

### 4.4. Ablation Study

The proposed method uses multiple strategies to deal with the fall recognition problem. To study the effectiveness of each strategy, we conducted several ablation experiments on the NTU dataset. The following is the main content.

**Sub-Graph Division and Sub-Graph Attention.** In our proposed method, sub-graphs are introduced to distinguish the various parts of the fall. To demonstrate the effectiveness of sub-graph division and sub-graph attention on fall recognition, we conducted several sets of comparison experiments on the collected NTU dataset (six types), including three groups, i.e., with only sub-graph division, with only sub-graph attention, and with both of them. As shown in Table 4, the results show that both sub-graph division and sub-graph attention in the collected NTU fall dataset effectively improve the recognition accuracy, effectively gather the features in the fall behavior, and fully explore the differences in the movement and change magnitude of each part of the human body during the fall.

**The Importance of MTCN.** In order to compare the effectiveness of the multi-scale temporal convolution network (MTCN), we conducted combined experiments on temporal convolution of different scale sizes. To evaluate the generalization ability of the model, the experiments were conducted on the NTU X-sub dataset instead of the collected fall dataset. As shown in Table 5, the proposed method is the least effective when the MTCN module is not used, reaching 85.7%. When the temporal convolution with a single convolutional kernel is taken, the recognition accuracy reaches 87.5%, 87.6%, and 87.2%, respectively. When the convolution kernels are combined two by two, the recognition accuracy reaches 88.9%, 88.4%, and 88.7%, respectively. When three scales are adopted, the accuracy reaches 89.8%. The results show that multi-scale temporal convolution is beneficial to extract richer temporal context information, which in turn improves the feature representation capability of the model and allows better modeling of temporal sequences.

### 4.5. Visualization

To better understand the effect of sub-graph attention, we visualized and analyzed the feature map of a fall behavior sequence after sub-graph attention. It can be observed in Figure 6 that, by introducing sub-graph attention, spatial-temporal features are more clustered at certain nodes and parts of the fall.

### 4.6. Time and Memory Cost Analysis

We conducted the experiments on a platform of a single GTX 2080Ti GPU. We also evaluated the training time, the execution speed, and the number of model parameters. As shown in Table 6, the training duration on the collected NTU, the skeleton of URFD, and UP-Fall are about 2 h, 4.5 h, and 7.5 h, respectively. Moreover, the execution speed on three datasets is 31 fp/s, 32 fp/s, and 30 fp/s, respectively. Meanwhile, we calculated the number of parameters of the model, which contains only 1.2 M. Compared with other methods, the fall recognition method based on vision is more convenient and faster. We obtain the human skeleton using OpenPose, and leverage GCN to model the spatial-temporal features of the human skeleton. This proves to be accurate and convenient. To some extent, the proposed method not only reaches high accuracy, but also has advantages of being simple, lightweight, and having a low cost. This means that our proposed method can be used for real-time detection and applied to cloud devices.

## 5. Conclusions

In this research, a lightweight subgraph-based deep learning approach for fall recognition is proposed. The skeleton information of the human body is extracted using OpenPose, while sub-graph division and sub-graph attention modules are introduced to add a larger perceptual field while maintaining its lightweight characteristics. A multi-scale temporal convolution module is also designed to extract and fuse multi-scale temporal features, which enriches the feature representation and improves the performance of fall recognition. The proposed method is evaluated on a partial fall dataset collected in NTU and on two public datasets. The experimental results and ablation study demonstrate that our method can achieve accurate fall recognition and outperform existing methods. The visual experiment intuitively proves the effectiveness of the sub-graph attention module. Besides, the analysis of time and memory cost shows that the proposed method can be applied to real-time detection and rapid response to fall events.

In the future, our research will continue utilizing 3D skeleton information of the human body to enhance the representation of falls. Besides, the proposed method may not only play a role in fall events, but probably also work for some actions with large changes, which is the direction of follow-up research.

## Figures and Tables

**Figure 1 sensors-22-05482-f001:**
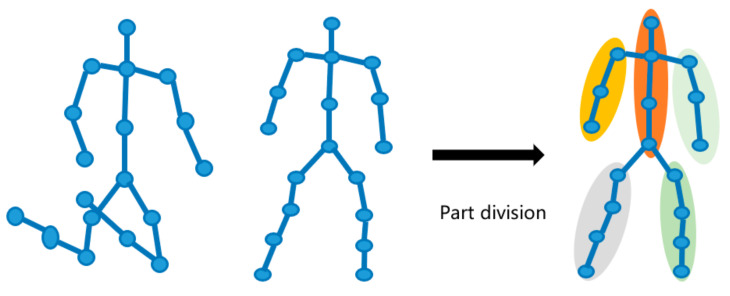
Schematic diagram of a fall and sub-graph division.

**Figure 2 sensors-22-05482-f002:**
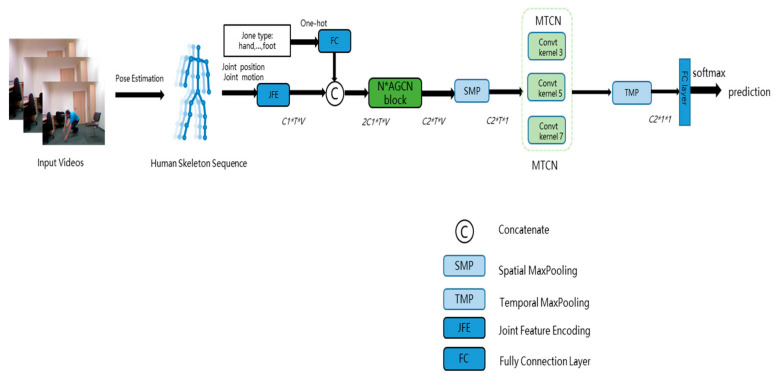
Illustration of the proposed method. It consists of skeleton extraction, a spatial module, and a temporal module.

**Figure 3 sensors-22-05482-f003:**
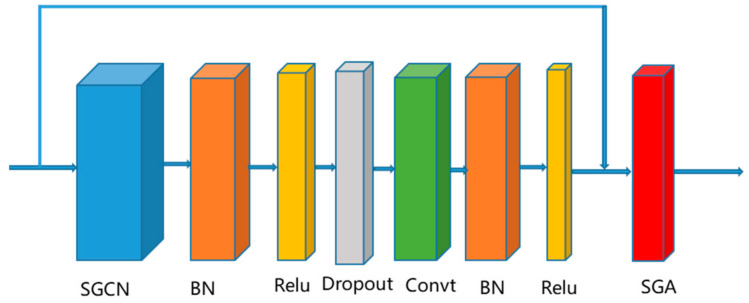
Composition of the AGCN. It consists of SGCN, a standard temporal convolution layer, and a sub-graph attention.

**Figure 4 sensors-22-05482-f004:**
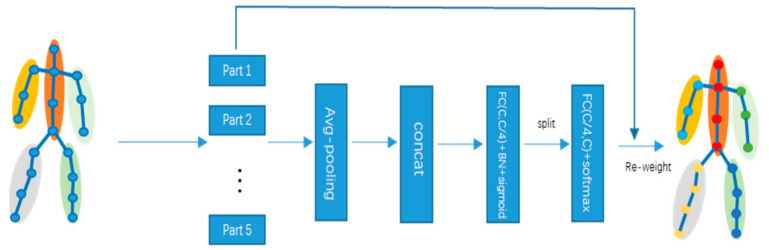
Illustration of the sub-graph attention module.

**Figure 5 sensors-22-05482-f005:**
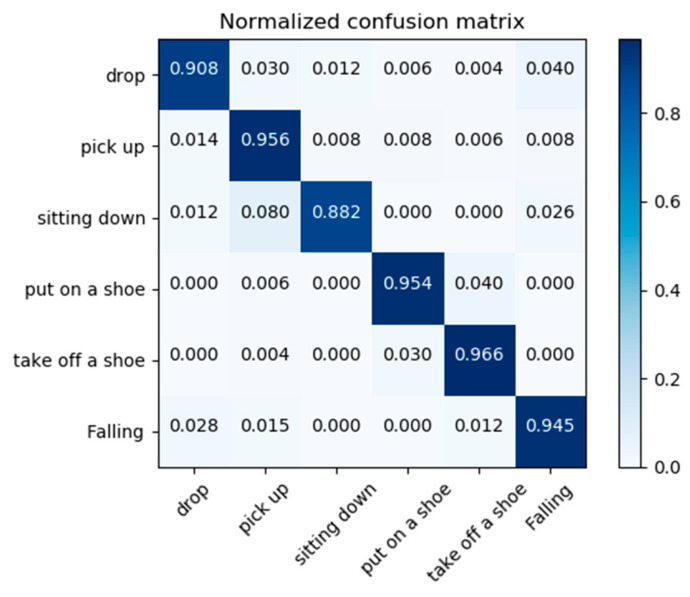
Confusion matrix for falls and similar actions on the collected NTU dataset.

**Figure 6 sensors-22-05482-f006:**
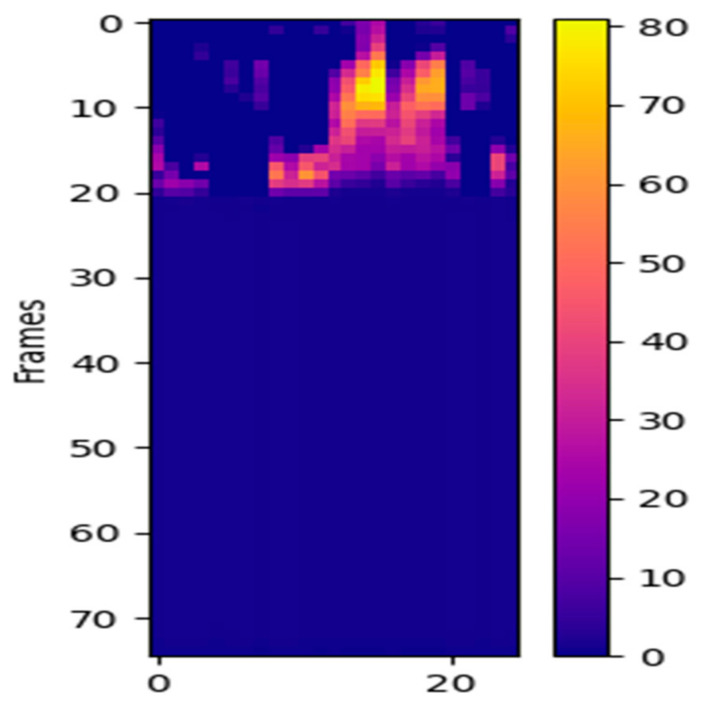
Sub-graph attention visualization feature map of a fall event.

**Table 1 sensors-22-05482-t001:** Comparison of methods on the UR Fall dataset (%).

Method	Sensitivity	Specificity	Accuracy
AR-FD [30]	98.0	89.4	94.0
MEWMA-FD [31]	100	94.9	96.6
Shi-Tomasi-FD [32]	96.7	-	95.7
CNN-FD [33]	100	92.0	95.0
CNN-LSTM-FD [34]	91.4	-	-
Proposed method	98.5	96.0	97.0

**Table 2 sensors-22-05482-t002:** Comparison of methods on the UP-Fall detection dataset (%).

Method	Sensitivity	Specificity	Accuracy
CNN + cam1 [35]	97.72	81.58	95.24
CNN + cam2 [35]	95.57	79.67	94.78
RF [7]	14.48	92.9	32.33
SVM [7]	14.30	92.97	34.40
MLP [7]	10.59	92.21	27.08
KNN [7]	15.54	93.09	34.03
CNN [7]	71.3	99.5	95.1
CNN [36]	99.5	83.08	95.64
RF + SVM + MLP + KNN [37]	96.80	99.11	98.59
CNN + LSTM [38]	94.37	98.96	98.59
Proposed method	95.43	99.12	98.85

**Table 3 sensors-22-05482-t003:** The experimental calculation results on the NTU dataset (six types of actions) (%).

	Sensitivity	Specificity	Accuracy
Result	97.5	89.6	94.5

**Table 4 sensors-22-05482-t004:** Performance of the proposed method on the collected NTU dataset (%).

Setting	NTU	
	X-Sub	X-View
Sub-Graph Division	90.3	92.8
Sub-Graph Attention	90.2	92.5
Sub-Graph Division + Sub-Graph Attention	**92.3**	**96.1**

**Table 5 sensors-22-05482-t005:** Comparison of temporal convolution with different scales (%).

Kernel Size			X-Sub
3	5	7	
			85.7
			87.5
√	√		87.6
		√	87.2
√	√		88.9
	√	√	88.4
√		√	88.7
√	√	√	89.8

**Table 6 sensors-22-05482-t006:** Time and memory cost analysis on the three datasets.

Dataset	Training Time (h)	Speed (fp/s)
collected NTU dataset	2	31
URFD	4.5	32
UP-Fall	7.5	30

## Data Availability

Not applicable.

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
