# Peer review of "A Lightweight Subgraph-Based Deep Learning Approach for Fall Recognition"

_sensors, 2022, doi:10.3390/s22155482_

Round 1

Reviewer 1 Report

In this article, the authors propose a novel method to detect falls of Ederly. They use a skeleton segmentation to separate the complexity of DL model in order to improve the precision/accuray of the results.

------------------------

Abstract

------------

- It seems word are missing: 

 L. 11 - the body's center of gravity moves from high to low ---> position?

 L. 19 - while maintaining IT lightweight 

1. Introduction

--------------------

- The first sentence is abnormally long!

- L.34 ...aged 65 or over HAS reacheD 727 ...

- L.35  In 2020, the proportion of the five elderly people --> What five means?

- L.49 ... Therefore, designing a lightweight and fast vision-49 based method for automatic fall recognition is an important task.   Why it is important compare to other technics ? more arguments.

- L. 51-63 Good explanation of the difficulties.

- L. 78 Firstly, we use the OpenPose skeleton extraction algorithm to obtain 78 the skeleton data of human body  Lake of ref. on OpenPose nor the type of algorithm.

- L 94 …

A total of six categories of falls and behaviors similar to falls are collected in the NTU dataset, and skeleton extraction was performed using OpenPose on two publicly available fall behavior datasets

--> Which dataset is used? Even if the datasets are presented in Sect. 4, it could be good to refer either the papers or the section in which the explanation is given.

2. Related work 

--------------------

-       L 101 – 114 : Could be good to have a glossary for the acronyms (RGB, CNN,….) for reader not specifically in the precise domain.

-       Sect. 2.3 . The related work could extended with the Wearable sensors especially because the is a lot of papers using this technics for Fall detection.

3. Method

-------------- 

-       Figure 3. doesn’t bring much information.

-       L.179 …An ordinary 1*1 convolutional layer –> Why 1*1? please argue! 

-       and a Relu layer are used to encode  the input features to obtain the high-dimensional expressions Jt,iR?1 and mt,iR?1,  and here ?1=16.  ïƒ  Why 16? Please argue! 

-       L.197 : Typo : , , 

-       L.235 . Where M denotes the maximum number of nodes in each sub-graph, and parts less  than M are filled with 0. Each part is stitched together after averaging pooling at the frame level : Why average pooling? To reduce the number of parameter? Argue!

4. experiment 

-------------------

-       L. 276 : IMPORTANT - The X-view takes the ones collected by camera 2 and 3 as the training set (37,920 samples) and camera 1 as the test set (18,960 276 samples).  If there is a separation of the data on different cameras, we have to be sure that the data distribution and the data characteristics are exactly the same otherwise the result will be biased!
Please explain!

-       L.287 : In this paper, we only use RGB images of 30 fall sequences and 30 normal activity sequences.  Why using 50/50 distribution knowing that in real life there will be more normal life sequence than fall. Pleas explain!

-       L. 290 : The UP-Fall datasets are collected from 17 young healthy subjects -> ok for the article but explain/mention clearly that this specificity doesn’t reflect the reality of a fall made by an Elderly.

-       L 313 : The training time was set to 60 epochs with an initial learning rate of 0.1 decreasing to the original 0.1: ---> Are you sure of the original value???

-       Figure 6 : Why the Sitting down is so high : please explain.

-       L. 375 : Sub-graph Division and SGA : Why one full name vs on acronym?

The conclusion could be a bit more developed.

Overall comments:

--------------------------

Few points can be improved.

-       Definitions of the acronym

-       Grammar and sentences (sometimes word are missing)

-       Unit in plain text.

-       Explain more the choice of architecture, parameters, etc.

Basically, why did you choose this type of design?

Reviewer 2 Report

The authors propose a lightweight subgraph-based deep learning approach for fall recognition. They claim their method outperforms existing ones on two public datasets.

I find the research interesting, but I wonder if this approach could be applied to some other actions, not only falls. The improvement is marginal over existing algorithms, and it raises the question of the need of such a complex approach for such a small gain. It would be nice if you elaborate about the suitability of your approach to the general problem of action recognition. Do you plan to release the code as open source?

NTU citation first appear too late, in line 267. Please add it when NTU is first named.

Line 35: what do you mean by "the proportion of the five elderly people"?

Lines 36 and 37: and outcome of falls is cardiovascular disease?

Line 103: please define RNN, CNN and GCN the first time they appear.

Line 278: what is NTU 60?

What do you mean by "scheme" in lines 339, 342, 346 and 349? "Method" or "approach"? Ah, I see that you took that from "A Novel Vision-Based Fall Detection Scheme Using Keypoints of Human Skeleton with Long Short-Term Memory Network", where the authors call everything "scheme", even their approach.

There are some [J] and [C]// in the bibliography which should not be there.

Please cite the OpenPose paper:

Cao, Zhe, Tomas Simon, Shih-En Wei, and Yaser Sheikh. "Realtime multi-person 2d pose estimation using part affinity fields." In Proceedings of the IEEE conference on computer vision and pattern recognition, pp. 7291-7299. 2017.

It is possible to improve the readilibity of the paper after careful revision. I suggest, for example, to rewrite the abstract. 
